# Assessment of Nutrient Levels Provided by General Hospital Patient Menus: A Cross-Sectional Study Carried Out in the Region of Murcia (Spain)

**DOI:** 10.3390/healthcare11162304

**Published:** 2023-08-15

**Authors:** Pablo Barcina-Pérez, Carmen Lucas-Abellán, Oriol Abellán-Aynés, María Teresa Mercader-Ros, Desirée Victoria-Montesinos, Pilar Hernández-Sánchez, Ana Serrano-Martínez

**Affiliations:** 1Faculty of Pharmacy and Nutrition, UCAM Universidad Católica San Antonio de Murcia, 30107 Murcia, Spain; pbarcina@ucam.edu (P.B.-P.); clucas@ucam.edu (C.L.-A.); mtmercader@ucam.edu (M.T.M.-R.); aserrano@ucam.edu (A.S.-M.); 2Faculty of Sport, UCAM Universidad Católica San Antonio de Murcia, 30107 Murcia, Spain; oabellan@ucam.edu

**Keywords:** hospital malnutrition, food service, hospital, nutritional requirements

## Abstract

Disease-related malnutrition remains a health problem with a high prevalence that increases the risk of poor patient outcomes, in addition to an elevation of healthcare costs. The aim of this study was to evaluate the nutritional quality of the menus at Ribera Molina Hospital, including their adequacy regarding recommended daily nutrient intakes and the agreement with the theoretical nutrition information provided by catering. The mean levels of energy, macronutrients, vitamins, and minerals provided by the basal, diabetic, and soft diets were calculated through the weighing of plated food served throughout the first 14 days of February 2020. A nutritional overestimation was seen in the nutrition information provided by the catering compared to the values derived from weighing foods (*p* < 0.01). Despite this, the nutritional content calculated by weighing satisfied the energy and protein requirements of 203 hospitalized patients previously studied in the internal medicine area of the hospital. The mean age of these patients was 62 years, and the main causes of admission were lung, cardiovascular, renal, and digestive diseases. There seems to be an insufficient amount of vitamins E and D, as well as magnesium, on all the menus. A possible insufficient amount of calcium, potassium, zinc, and copper was observed in some of the menus studied. It is necessary to update the hospital prescription manual so the nutritional contents of the diets are accurate and based on the weighted and calculated values to improve the adequacy of diets prescribed to patients.

## 1. Introduction

Disease-related malnutrition (DRM) represents one of the most prevalent conditions in hospital care. Although work continues to standardize criteria for screening and diagnosis, and there is still some disparity in results, it is known that this problem affects more than 30 million people in Europe, and this represents an economic cost of about EUR 170 billion every year [1]. DRM is associated with adverse clinical outcomes, including delayed wound healing, increased length of hospital stay, increased readmission rates, and, as a result, higher healthcare costs [2,3]. In Spain, the prevalence of DRM among patients admitted to hospitals is estimated to be between 23% and 37%, according to two of the main studies carried out at the national level [4,5].

DRM is partly caused by the increased energy and protein requirements associated with acute or chronic diseases, in addition to the reduced dietary intake due to nutrition-impacting symptoms associated with the disease [2]. In addition, metabolic stress generated in pathological processes can increase the requirements of micronutrients, especially water-soluble vitamins that act as coenzymes due to the increased activity of the different metabolic pathways [2,6,7]. This can lead to an increase in reactive oxygen species (ROS), which increases the need for antioxidants, especially vitamins E and C [8,9,10]. According to government data, it is observed that between 25% and 75% of the population in Europe and the United States have a dietary intake below the Reference Daily Intake (RDI) for one or more micronutrients [11]. Specifically, insufficient intakes have been observed for calciferol (vitamin D), alpha-tocopherol (vitamin E), retinoids (vitamin A), ascorbic acid (vitamin C), folates (vitamin B9), zinc (Zn), selenium (Se), and magnesium (Mg), regardless of age [11].

Due to these facts, it is important for hospital menu planning to ensure that menus provide adequate energy, macronutrients, vitamins, and minerals to meet the nutritional needs of patients during a hospital stay [12]. Hospital diets should guarantee an adequate supply of nutrients for patients; however, there is a lack of official regulations that standardize both the names of the diets and their nutritional aspects, which has led to hospitals being able to develop their own menus and diet names; and in some cases, there are not even internal guidelines [13]. It is important to assess the nutritional adequacy of hospital diets in order to guarantee optimal nutrient intake. This may be performed by evaluating the nutrition information provided by the kitchen services, weighing foods and calculating nutrient contents from the manufacturers’ label, or chemical analysis [12,14,15].

Currently, few studies in Spain have evaluated the nutritional quality of menus for hospital patients [15]. However, such work has been performed in other countries [16,17]. While guidelines for the nutritional content of hospital menus exist [18], few studies have analyzed the degree to which hospital menus meet these guidelines. To do this, a greater degree of certainty about the actual nutritional content of hospital menus is required [12,13]. 

The present work was informed by a previous study on DRM in Ribera Molina Hospital [19]. The aim of the current study was to evaluate the nutritional quality of the menus most frequently served at this hospital. The mean levels of energy, macronutrients, vitamins, and minerals provided by the basal, diabetic, and soft diets were calculated by weighing. The secondary aim was to determine the degree of compliance of these menus with the RDI, as well as the degree of agreement between the nutrition information provided by catering and the data derived from the food weighing study.

## 2. Materials and Methods

This research was undertaken at the Ribera Molina Hospital and the Catholic University of Murcia. The food service at Ribera Molina Hospital is divided into four meals: breakfast at 8:00 a.m., lunch at 1:00 p.m., afternoon snack at 5:00 p.m., and dinner at 8:00 p.m. Seven types of diets are used: basal, diabetic, soft, turmix (texture-modified diet), astringent (low in insoluble fiber and fat, free of lactose, irritating or flatulent foods), astringent turmix, and low-fat. All the menus have a rotation of 14 days and two seasons, i.e., 14 days in winter and 14 days in summer, which rotate systematically.

The basal diet provides approximately 2200–2400 kcal (50–55% carbohydrates, 30–35% lipids, and 10–20% proteins), as per the catering service information. It is estimated that the micronutrients provided by the basal diet are those that, at least, meet the recommended daily intakes. Therapeutic diets are made from modifications of the basal menu. In the diabetic diet, the contribution of carbohydrates is reduced, and in the soft diet, the fiber and fat content are reduced.

By weighing the plated food, the contribution of energy, macronutrients, vitamins, and minerals provided by the menus corresponding to the basal, diabetic, and soft diets (Appendix A), which were served each of the 14 days of the sampling period, were evaluated. The menu corresponding to the low-fat diet was discarded because it was not served on a sufficient number of occasions during the 14 days that the study took place, and those with modified textures were discarded because of the technical impossibility of weighing the ingredients separately.

Food was weighed during plating at lunch and dinner during the two weeks of the first half of February 2020. At each plating, one of the prepared trays corresponding to each of the three menus under study was randomly selected. The Imetec Dolcevita 7786 ES4 scale, with a range of five kilograms (kg) and an accuracy of one gram (g), was used for weighing. Once all the weights of the foods had been obtained separately and drained, a nutritional assessment was carried out using Icns-medical software^®^ version 1.0. To establish more accurate comparisons, this software was also used to nutritionally analyze nutrition information provided by catering. In both evaluations, data on the breakfast and afternoon snacks provided by the catering service were introduced, which were divided according to each of the three types of diet analyzed in two-day slots throughout the week: one for Monday, Wednesday, and Friday, and another for Tuesday, Thursday, Saturday, and Sunday. 

To assess the adequacy of the different menus for the requirements of the patients, the nutritional values obtained in the weighted assessment were compared with the theoretical requirements of the patients assessed in our previous study who were assigned a basal, carbohydrate-controlled, or soft diet [19]. Two hundred and three patients were included in that study, of which 52.7% were women; the mean age was 62.4 years (SD, 20.3) and the mean body mass index (BMI) was 28.5 kg/m^2^ (SD, 6.0). They were all derived from the Public Health System and had similar sociodemographic characteristics. The main causes of admission were related to lung (32.2%), cardiovascular (18.8%), renal (14.4%), and digestive (13.4%) diseases, and less frequently pancreatic, urinary, musculoskeletal, metabolic, nervous, and immunological diseases. Their resting metabolic rate (RMR) was calculated according to recommendations of the American Society for Parenteral and Enteral Nutrition (ASPEN) [20], in which the Mifflin St. Jeor formula (Women = (10 × kg) + (6.25 × cm) − (5 × age) − 161; Men = (10 × kg) + (6.25 × cm) − (5 × age) + 5) is proposed as the most accurate when estimating the RMR for non-critical patients with or without obesity, always using the actual unadjusted weight [21]. A factor of 1.3 was applied to the RMR for slightly hypermetabolic patients not bed-confined 100% of the admission time, such as those suffering from pneumonia, exacerbation of chronic obstructive pulmonary disease, infection, peritonitis, and fever [22]. Regarding protein requirements, the range of 1.2–1.5 g/kg/day was established [23], according to the actual weight for body mass index (BMI) < 27 kg/m^2^ and the adjusted ideal weight (AIW) [24] for a BMI ≥ 27 kg/m^2^ (AIW = [(Actual weight − IW) × 0.25] + IW). For ideal weight (IW) the Lorentz formula was used (IW = Height (cm) − 100 − [(height (cm) − 150)/k] K men = 4; K women = 2) [25,26].

Theoretical micronutrient requirements were estimated based on the consensus of the RDI for the Spanish population and the legislation related to nutritional labeling [27,28].

For the statistical analysis, SPSS for Windows (version 25.0, Armonk, NY, USA: IBM Corp) was used to study the differences between the data provided by the catering and the nutritional data obtained by weighing, as well as the data obtained by weighing and the estimated nutritional requirements for the patients. The data were expressed as mean and standard deviation (SD) and were compared using Student’s *t*-test with a significance value set at *p* < 0.05.

## 3. Results

The nutritional assessment by weighing the food corresponding to the menus studied was carried out during a complete 14-day rotation during the winter season. The results obtained through the nutritional software show that the basal diet provides a greater amount of nutrients and energy than the rest of the diets, except in the protein section, where it is the soft diet that offers a greater protein contribution (Table 1 and Table 2).

In general, there was a nutritional overestimation in the information provided by the catering compared to the calculations derived from the weighing (Table 1). 

Although no statistically significant differences were observed between the different menus, a statistically significant difference (*p* < 0.01) was observed between the average values, corresponding to nutrition information provided by catering and that obtained in the assessment by weighing the food in all cases (Table 3).

This difference shows an overestimation of the nutritional values described in the nutrition information provided by catering, both in terms of energy and grams of carbohydrates, fat, and protein (Table 3).

Although an overestimation of the nutritional values was observed in the nutrition information provided by catering, the nutritional values calculated by weighing satisfied the estimated energy and protein requirements of the patients assigned basal and soft diets. However, the energy intake of the diabetic diet was, on average, slightly insufficient to satisfy the requirements of the patients to whom it was assigned (Table 1).

From the study of the percentage distribution of macronutrients according to the total caloric value (TCV), it can be observed that the fat content is below the maximum recommended 30%. It is also observed that the maximum recommended percentage of 15% protein is slightly exceeded in all of them, and the carbohydrate amount is within the recommended range, between 45 and 60% [29].

Regarding the contribution of micronutrients corresponding to the evaluation by weighing the basal, diabetic, and soft diets (Table 4), it can be seen that in all three cases, the RDIs established for the healthy population were reached without exceeding the upper limit of thiamine (vitamin B1), riboflavin (vitamin B2), riboflavin (vitamin B3), pyridoxine (vitamin B6), folate (vitamin B9), cobalamin (vitamin B12), retinoids (vitamin A), ascorbic acid (vitamin C), phylloquinone (vitamin K), iron (Fe), phosphorus (P), sodium (Na), and selenium (Se). However, there was an insufficient amount of alpha-tocopherol (vitamin E) to meet the RDI (15 mg) in all the menus evaluated by weighing, providing only 62.0%, 69.3%, and 42.8% of the RDI in the basal, diabetic, and soft diets, respectively.

Similarly, an insufficient amount of calciferol (vitamin D) was observed in all the menus evaluated by weighing, providing only 35.3%, 42.1%, and 55.0% of the RDI in the basal, diabetic, and soft diets, respectively. Regarding calcium (Ca), it was observed that the diabetic diet provided only 83.0% of the RDI.

All menus evaluated by weighing had insufficient magnesium (Mg) (Table 5), providing only 86.7%, 78.4%, and 70.4% of the RDI in the basal, diabetic, and soft diets, respectively. All the menus reached the RDI for potassium (K) except the soft diet, which provided only 86.6% of the RDI. There was an insufficient amount of zinc (Zn) in the diabetic and soft diets, providing 86.6% and 85.4% of the RDI, respectively. The basal diet had an acceptable copper (Cu) intake since it exceeded 90% of the RDI; however, the diabetic and soft diets had an inadequate amount of 55.1% and 59.3% of the RDI, respectively.

Table 6 compares the theoretical nutrition information provided by catering, the values obtained by weighing, and the nutritional objectives for the Spanish population [29,30]. There was an excess in the phosphorus nutrient provision, which created a certain imbalance in the ratio of Ca (Ca/P), which should be 1.3/1. The ratio between the number of vitamins B1, B2, and B3 per 1000 Kcal in all cases exceeds the minimum recommendations of 0.4, 0.6, and 6.6 mg/1000 Kcal, respectively. The ratio between the amount of vitamin B6 and protein is maintained at the minimum recommendation of 0.02 mg per gram of protein. It was observed that the amount of vitamin E per gram of polyunsaturated fatty acid was higher than the minimum recommendation of 0.4 mg. 

The fats in all the menus evaluated by weighing the food presented a good-quality nutritional proportion, and the ratio between polyunsaturated fatty acids and saturated fatty acids is higher than the recommended 0.5 in all the cases. 

The ratio between the sum of polyunsaturated and monounsaturated fatty acids compared to saturated fatty acids was higher than the recommended two in all cases.

## 4. Discussion

This study evaluated the nutritional content of hospital diets by weighing and compared the diets to the nutritional information provided by the hospital catering and the RDI of a cohort of patients previously studied. 

The nutritional contribution calculated by weighing satisfied the energy and protein requirements of the patients. The contribution of protein to total energy was between 17% and 20% of the TCV, which exceeded the range recommended in the nutritional objectives for the Spanish population (NO) of between 10% and 15% of the TCV [30]. However, this is not likely to be a problem, as hospital patients often have increased protein requirements. If the provision of carbohydrates from the hospital diets calculated by weighing food is observed, it can be seen that the proportions of this macronutrient meet the range of 50–60% over the TCV established in the NO for the Spanish population [30]. Regarding fat provision, the assessment by weighing showed a correct distribution related to the TCV, being below 30% in all cases, as recommended in the NO for the Spanish population. Likewise, a correct balance between monounsaturated, polyunsaturated, and saturated fatty acids was observed [30].

This study found that despite the overestimation of energy and protein contents of hospital diets in the hospital’s catering information, the weighted assessment showed the diets still contained adequate energy and protein to meet the needs of patients in most cases. This is important, as meeting the energy and protein requirements of patients during a hospitalization is critical to their recovery [19,30,31]. Another study found that while the basal/regular diet met patient requirements, many of the therapeutic diets did not [32].

If this same information was compared with the standards of the specifications established by the Regional Ministry of Health of the Region of Murcia, it can be seen that with the exception of the diet for diabetic patients, the rest would exceed the maximum Kcal established at 2000 Kcal per day [33].

While our study found energy and protein provision to patients in hospital diets was adequate, it is not guaranteed that patients will eat all of their meals. In fact, studies show that patients eat on average only 60% of their meals, so it is important that adequate nutrition care is implemented for those at nutritional risk [34,35].

This study found deficiencies regarding the RDIs in the provision of calciferol (vitamin D), alpha-tocopherol (vitamin E), zinc (Zn), magnesium (Mg), and copper (Cu). The micronutrient needs of hospitalized patients may be even higher than those proposed in the RDIs for the general population [11], especially in patients affected by pulmonary, cardiovascular, digestive, and renal pathology (who also experience high rates of DRM), with a probable increase derived from the metabolic stress associated with their pathology [10,36]. Given this, there may be an even higher discrepancy between micronutrient provision and actual requirements for these patients.

Metabolic stress can increase their requirements, especially for water-soluble vitamins that act as coenzymes due to the increased activity of different metabolic pathways. This may generate an increase in ROS, which would increase the need for antioxidants, especially vitamins E and C. In addition, vitamin status can be affected by poor distribution of vitamins throughout the body due to a drop in the concentration of carrier proteins due to losses through body fluids, derived from treatments, such as dialysis, or pathologies, such as diarrhea [8,9,10]. For example, a 50% reduction in serum vitamin C levels is estimated during infection, and it is believed that supplementation with doses above the RDI (100–200 mg/day) may serve as a preventive factor against infection, and even as part of treatment for infection, with higher intakes alleviating the increase in metabolic demand during the inflammatory response [11]. Vitamin E was one of the deficient micronutrients in menus in the present study, with mean daily values ranging between 6.3 mg and 10.4 mg, well below the RDI of 15 mg. Not only did hospital menus fail to meet the RDI, but it is also likely that some of the patients admitted to the hospital presented deficient states [11]. Research suggests that the use of vitamin E supplements above the RDI could help the function of the T cells of the immune system, as well as increase the efficacy of vaccination, which is diminished during the aging process [11].

Another micronutrient of clinical relevance is calciferol (vitamin D). Some studies suggest that current RDIs may be insufficient for optimal bone health in the elderly population and the proper functioning of the immune system [11,37]. Data have recently been published on the possible increased risk of developing COVID-19 in persons with vitamin D deficiency or suboptimal vitamin D status. This issue could be worsened by the processes of confinement in which exposure to sunlight has been reduced so that treatment with vitamin D supplementation, at least in people with deficiency states, could reduce the risk of developing COVID-19 and could improve the clinical course during the disease once it has been contracted.

Another mineral that is being studied in this context is Zinc (Zn). In this study, it was observed that zinc provision did not reach the RDI (10 mg) in the diabetic and soft diets. There seems to be an insufficient zinc intake in some industrialized countries, so the effects of a deficiency or suboptimal state could be worsened if, during the hospital stay, not even the RDI is reached [11].

This study is not without its limitations, which warrant careful consideration. While the authors deem this line of research crucial, the results presented should be interpreted with caution due to the sample size of the study and potential deviations in the indirect estimation of patients’ nutritional requirements.

An additional significant limitation acknowledged by the authors is the absence of data regarding non-hospital-provided food intake, as well as the amount of hospital-provided food that was left uneaten.

The necessity for further investigation is clear. Moreover, robust scientific evidence is required to validate the application of micronutrient supplementation in a clinical setting as a preventative measure against certain diseases and as an adjunct in medical treatment. Research is needed to establish optimal micronutrient intake levels that can better inform dietary guidelines for the public and hospital menu planning.

## 5. Conclusions

This study found discrepancies between the nutritional information on menus provided by the hospital catering and the nutritional values found by weighing the foods. Hence, it is necessary to update the hospital’s catering information to ensure greater accuracy in estimating patients’ nutrient intakes. Although the energy and protein provision of the hospital menus met the nutritional objectives for the Spanish population, our assessment detected an insufficient provision of vitamin E, vitamin D, zinc, and magnesium. Therefore, when updating the menus, special attention should be paid to these micronutrients. This study highlights important considerations for the planning of hospital menus to meet patients’ nutrient requirements, especially for micronutrients. Future research should explore the adequacy of hospital diets for patients (including energy, macronutrients, and micronutrients) in different settings around the world.

## Figures and Tables

**Table 1 healthcare-11-02304-t001:** Comparison of diets’ recommended, theoretical (per catering information), and actual (weighed) daily energy and protein contents.

	Mean Estimated Patient Requirement	Nutrition Information Provided by Catering	Weighted Assessment
**Basal diet**			
**Energy (kcal)**	1885 (SD, 301)	3043 (SD, 295)	2431 (SD, 208) *
**Protein (g)**	78.4 (SD, 9.9)	143.4 (SD, 20.5)	107.1 (SD, 19.8) *
**Diabetic diet**			
**Energy (kcal)**	1863 (SD, 279)	2440 (SD, 227)	1790 (SD, 136) *
**Protein (g)**	79.5 (SD, 9.8)	130.0 (SD, 21,4)	91.3 (SD, 15.1) *
**Soft diet**			
**Energy (kcal)**	1853 (SD, 365.56)	2592 (SD, 225)	2123 (SD, 279) *
**Protein (g)**	77.5 (SD, 12.6)	136.48 (SD, 18,2)	108.6 (SD, 15.9) *

Kilocalories (Kcal); grams (g); standard deviation (SD). ^(^*^)^ Difference between theoretical values provided by the catering information and valuation by weighing *p* < 0.01.

**Table 2 healthcare-11-02304-t002:** The daily nutritional content of hospital diets analyzed by weighing.

	Type of Diet	Mean	SD
**Energy (Kcal)**	Basal diet	2431	208
	Diabetic diet	1790	136
	Soft diet	2123	279
**Carbohydrates (g)**	Basal diet	321.1	25.7
	Diabetic diet	215.1	26.1
	Soft diet	278.2	20.9
**Protein (g)**	Basal diet	107.1	19.8
	Diabetic diet	91.3	15.1
	Soft diet	108.6	15.9
**Fat (g)**	Basal diet	73.0	16.7
	Diabetic diet	57.5	13.1
	Soft diet	60.5	23.2
**Fiber (g)**	Basal diet	30.6	5.4
	Diabetic diet	24.0	4.5
	Soft diet	15.8	6.2

Kilocalories (Kcal); grams (g); standard deviation (SD).

**Table 3 healthcare-11-02304-t003:** Nutritional differences between the information provided by the catering and the weighted assessment.

	Difference	SD	CI 95%	
			Lower	Upper
**Energy (Kcal)**				
Basal weighed-Basal catering	−612 ^(^*^)^	297	−783	−440
Diabetic weighed-Diabetic catering	−649 ^(^*^)^	256	−797	−502
Soft weighed-Soft catering	−468 ^(^*^)^	300	−642	−295
**Carbohydrates (g)**				
Basal weighed-Basal catering	−40.4 ^(^*^)^	31.6	−58.7	−22.2
Diabetic weighed-Diabetic catering	−49.8 ^(^*^)^	25.6	−64.6	−35.1
Soft weighed-Soft catering	−32.0 ^(^*^)^	16.7	−41.6	−22.4
**Proteins (g)**				
Basal weighed-Basal catering	−36.3 ^(^*^)^	20.3	−48.0	−24.6
Diabetic weighed-Diabetic catering	−38.7 ^(^*^)^	17.2	−48.6	−28.8
Soft weighed-Soft catering	−28.0 ^(^*^)^	16.8	−37.6	−18.2
**Fat (g)**				
Basal weighed-Basal catering	−31.4 ^(^*^)^	23.7	−45.1	−17.6
Diabetic weighed-Diabetic catering	−30.0 ^(^*^)^	21.3	−42.3	−17.8
Soft weighed-Soft catering	−22.0 ^(^*^)^	23.4	−35.4	−8.4
**Fiber (g)**				
Basal weighed-Basal catering	−11.1 ^(^*^)^	9.3	−16.5	−5.8
Diabetic weighed-Diabetic catering	−12.3 ^(^*^)^	8.5	−17.2	−7.5
Soft weighed-Soft catering	−5.9 ^(^*^)^	6.2	−9.5	−2.3

Kcal, kilocalories; g, grams; SD, standard deviation; CI, confidence interval. ^(^*^)^
*p* < 0.01.

**Table 4 healthcare-11-02304-t004:** Vitamin assessment.

Weighted Assessment	Mean (SD)	RDI
**Basal diet**		
**Vitamin B1 (mg)**	1.9 (0.5)	1.2
**Vitamin B2 (mg)**	1.9 (0.3)	1.2
**Vitamin B3 (mg)**	28.0 (4.7)	15
**Vitamin B6 (mg)**	1.8 (0.5)	1.3
**Vitamin B9 (μg)**	655.7 (130.4)	400
**Vitamin B12 (μg)**	5.4 (4.3)	2.4
**Vitamin A (μg)**	1079.4 (397.2)	800
**Vitamin C (mg)**	216.4 (107.0)	80
**Vitamin E (mg)**	9.3 (2.8)	15
**Vitamin D (IU)**	211.6 (70)	600
**Vitamin K (μg)**	348.0 (350.9)	90
**Diabetic diet**		
**Vitamin B1 (mg)**	1.6 (0.4)	1.2
**Vitamin B2 (mg)**	1.9 (0.3)	1.2
**Vitamin B3 (mg)**	21.7 (5.6)	15
**Vitamin B6 (mg)**	1.9 (0.5)	1.3
**Vitamin B9 (μg)**	440.3 (105.4)	400
**Vitamin B12 (μg)**	6.7 (5.2)	2.4
**Vitamin A (μg)**	1019.7 (344.2)	800
**Vitamin C (mg)**	289.6 (129.9)	80
**Vitamin E (mg)**	10.4 (3.2)	15
**Vitamin D (IU)**	252.7 (176.9)	600
**Vitamin K (μg)**	337.8 (263.2)	90
**Soft diet**		
**Vitamin B1 (mg)**	1.9 (0.2)	1.2
**Vitamin B2 (mg)**	2.0 (0.4)	1.2
**Vitamin B3 (mg)**	23.4 (6.7)	15
**Vitamin B6 (mg)**	1.9 (0.5)	1.3
**Vitamin B9 (μg)**	525.7 (110.9)	400
**Vitamin B12 (μg)**	7.5 (5.3)	2.4
**Vitamin A (μg)**	1089.3 (653.6)	800
**Vitamin C (mg)**	120.2 (244.3)	80
**Vitamin E (mg)**	6.4 (3.7)	15
**Vitamin D (IU)**	329.7 (292.6)	600
**Vitamin K (μg)**	173.3 (313.3)	90

SD: standard deviation, Kcal: kilocalories, mg: milligrams, μg: micrograms, IU: international units, RDI: recommended dietary intake.

**Table 5 healthcare-11-02304-t005:** Mineral assessment.

Weighted Assessment	Mean (SD)	RDI
**Basal diet**		
**Calcium (mg)**	1116.7 (164.3)	1100
**Iron Basal (mg)**	17.6 (3.0)	10
**Magnesium (mg)**	329.3 (60.1)	380
**Phosphorus (mg)**	1614.9 (254.1)	700
**Potassium (mg)**	4121.1 (855.2)	3510
**Sodium (mg)**	2541.3(475.4)	2300
**Zinc (mg)**	10.8 (2.6)	10
**Selenium (μg)**	134.4 (29.4)	55
**Copper (μg)**	824.4 (242.1)	900
**Diabetic diet**		
**Calcium (mg)**	910.5 (137.5)	1100
**Iron (mg)**	11.8 (2.5)	10
**Magnesium (mg)**	298.0 (65.5)	380
**Phosphorus (mg)**	1426.8 (220.2)	700
**Potassium (mg)**	3713.3 (763.8)	3510
**Sodium (mg)**	1862.3 (319.9)	2300
**Zinc (mg)**	8.7 (2.4)	10
**Selenium (μg)**	103.0 (29.0)	55
**Copper (μg)**	496.0 (298.8)	900
**Soft diet**		
**Calcium (mg)**	1258.6 (197.6)	1100
**Iron (mg)**	14.6 (2.3)	10
**Magnesium (mg)**	267.6 (53.1)	380
**Phosphorus (mg)**	1426 (203.9)	700
**Potassium (mg)**	3039.7 (620.7)	3510
**Sodium (mg)**	2909.6 (326.6)	2300
**Zinc (mg)**	8.5 (2.7)	10
**Selenium (μg)**	136.6 (39.1)	55
**Copper (μg)**	533.9 (185.4)	900

SD: standard deviation, mg: milligrams, μg: micrograms, RDI: recommended dietary intake.

**Table 6 healthcare-11-02304-t006:** Nutritional quality of the menus evaluated theoretically and by weighing with respect to the nutritional objectives.

	Spanish Nutritional Objectives	Weighing-Assessed Basal Diet	Basal Diet Assessed according to Catering	Weighing-Assessed Diabetic Diet	Diabetic Diet Assessed according to Catering	Weighing-Assessed Soft Diet	Soft Diet Assessed according to Catering
**% TCV Proteins**	10–15	17.60	18.90	20.40	21.30	20.50	21.10
**% TCV CH**	50–60	52.80	47.50	48.10	43.40	52.40	48.00
**% TCV Fat**	<30.00	27.00	30.90	28.90	32.30	25.70	28.60
**PUFAs/SFAs**	≥0.50	0.82	0.84	0.96	0.93	0.75	0.80
**(PUFAs + MUFAs)/SFAs**	≥2.00	2.74	3.22	3.11	3.47	2.11	2.80
**Calcium/Phosphorus**	1.31	0.69	0.61	0.64	0.56	0.88	0.77
**B1 mg/1000 Kcal**	0.40	0.77	0.94	0.88	0.93	0.91	0.91
**B2 mg/1000 Kcal**	0.60	0.79	0.84	1.03	0.90	0.93	0.89
**B3 mg/1000 Kcal**	6.60	11.50	13.17	12.13	13.77	11.03	12.81
**B6 mg/Proteins g**	>0.02	0.02	0.02	0.02	0.02	0.02	0.02
**Vit E mg/PUFAs g**	>0.40	0.73	0.92	0.86	1.01	0.58	0.67

TCV: total caloric value, CH: carbohydrates, PUFAs: polyunsaturated fatty acids, MUFAs: monounsaturated fatty acids, SFAs: saturated fatty acids Kcal: kilocalories mg: milligrams; g: grams; Vit: vitamin.

## Data Availability

Data are contained within the article.

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
