# Peer review of "Assessment of Nutrient Levels Provided by General Hospital Patient Menus: A Cross-Sectional Study Carried Out in the Region of Murcia (Spain)"

_healthcare, 2023, doi:10.3390/healthcare11162304_

Round 1
Reviewer 1 Report
Dear author(s),
The paper is good designed. However, micronutrients should be given. Also, is there a statistical difference between macronutrients according to diet type? This should also be provided. The abstract section will be detailed according to the results suggestions.
Also, what is the inpatient population in this hospital? information should be given about it. Because, although the macronutrients of diet types are calculated, they should be interpreted according to the inpatient profile. In this context, it is concluded whether the food served is sufficient or not.
Reviewer 2 Report
Thank you, dear editors, for the honor of being associated with the review of this article on the assessment of nutrient levels in the menus of general hospital patients. This study is important for several reasons: patients almost always have special nutritional needs, and a healthy, balanced diet is essential for healing and recovery. Such a study might help to identify gaps in the food offered in hospitals and to promote a healthy and adequate diet. As the authors point out, malnutrition linked to illness remains a health problem with a high prevalence that increases the risk of a more unfavorable clinical course in malnourished patients, in addition to increasing healthcare costs. The aim of their study was to assess the nutritional quality of the menus at the Ribera Molina Hospital and determine the degree of compliance with the recommended daily allowances and the theoretical information provided by the catering service.
Our main observations are set out below.
Title
We think that the authors should add the type and, if possible, the location of the study.
Abstract
This seems to be well structured and presents all the elements needed to understand the study.
Introduction
The introduction seems to us to set out the context, justification, presentation of the problem, and objectives of the study.
Materials and methods
We think that this subsection is incomplete. The authors do not clarify the choice of this hospital.
Results
We suggest adding the following to the title of Table 1: "Daily.....".
Discussion
We propose to begin the discussion by reviewing the results. As a reminder, the primary objective was to assess the nutritional quality of the meals frequently served in the hospital, and the secondary objective was to determine the extent to which these menus complied with the RDI, as well as the degree of agreement between the theoretical nutritional information provided in the hospital prescription data sheets and the data derived from the food weighing study.
Furthermore, the discussion does not address the main limitations of the study or the possibility of generalizing the results.
Reviewer 3 Report
Generally, the manuscript brought up the diet quality that can support the patient's condition. It is interesting. However, there are some issues that still can be improved and need to be confirmed by the authors.
Authors need to give more explanation about the diet types, for example, the details of food names or ingredients in one of the menu cycles. Basal diet is consisted of how many macronutrients and micronutrients are compared to the hospital standard menus.
and authors may need to add a brief explanation about the patients that not only consume the foods from hospital that may contribute to their health conditions.
